# RMIX: Learning Risk-Sensitive Policies for Cooperative Reinforcement Learning Agents

**Wei Qiu**[*1], **Xinrun Wang**[1], **Runsheng Yu**[2], **Xu He**[1], **Rundong Wang**[1],
**Bo An**[1], **Svetlana Obraztsova**[1], **Zinovi Rabinovich**[1]
[1]Nanyang Technological University, Singapore
[2]Hong Kong University of Science and Technology, Hong Kong

## Abstract

Current value-based multi-agent reinforcement learning methods optimize individual Q values to guide individuals' behaviours via centralized training with decentralized execution (CTDE). However, such expected, i.e., risk-neutral, Q value is not sufficient even with CTDE due to the randomness of rewards and the uncertainty in environments, which causes the failure of these methods to train coordinating agents in complex environments. To address these issues, we propose RMIX, a novel cooperative MARL method with the Conditional Value at Risk (CVaR) measure over the learned distributions of individuals' Q values. Specifically, we first learn the return distributions of individuals to analytically calculate CVaR for decentralized execution. Then, to handle the temporal nature of the stochastic outcomes during executions, we propose a dynamic risk level predictor for risk level tuning. Finally, we optimize the CVaR policies with CVaR values used to estimate the target in TD error during centralized training and the CVaR values are used as auxiliary local rewards to update the local distribution via Quantile Regression loss. Empirically, we show that our method outperforms many state-of-the-art methods on various multi-agent risk-sensitive navigation scenarios and challenging StarCraft II cooperative tasks, demonstrating enhanced coordination and revealing improved sample efficiency.

## 1 Introduction

Reinforcement learning (RL) has made remarkable advances in many domains, including arcade video games [28], complex continuous robot control [21] and the game of Go [40]. Recently, many researchers put their efforts to extend the RL methods into multi-agent systems (MASs), such as urban systems [41], coordination of robot swarms [16] and real-time strategy (RTS) video games [50]. Centralized training with decentralized execution (CTDE) [30] has drawn enormous attention via training policies of each agent with access to global trajectories in a centralized way and executing actions given only the local observations of each agent in a decentralized way. Empowered by CTDE, several multi-agent RL (MARL) methods, including value-based and policy gradient-based, are proposed [10, 43, 35, 42]. These MARL methods propose decomposition techniques to factorize the global Q value either by structural constraints or by estimating state-values or inter-agent weights to conduct the global Q value estimation [10, 43, 35, 42, 53, 54].

Despite the merits, most of these works focus on decomposing the global Q value into individual Q values with different constraints and network architectures, but ignore the fact that such expected, i.e., risk-neutral, Q value is not sufficient as optimistic actions executed by some agents can impede the team coordination such as imprudent actions in hostage rescue operations, which causes the failure of

---

[*]Correspondence to `qiuw0008@e.ntu.edu.sg`

these methods to train coordinating agents in complex environments. Specifically, these methods only learn the expected values over returns [35] and do not handle the high variance caused by events with extremely high/low rewards to agents but at small probabilities, which cause the inaccurate/insufficient estimations of the future returns. Therefore, instead of expected values, learning distributions of future returns, i.e., Q values, are more useful for agents to make decisions. Even further, given that the environment is nonstationary from the perspective of each agent, decision-making over the agent's return distribution takes events of potential return into account, which makes agents able to address uncertainties in the environment compared with simply taking the expected values for execution. However, current MARL methods do not extensively investigate these aspects.

Motivated by the previous reasons, we intend to extend the risk-sensitive RL [5, 18, 56, 3] ("Risk" refers to the uncertainty of future outcomes [8]) to MARL settings, where risk-sensitive RL optimizes policies with a risk measure, such as variance, power formula measure value at risk (VaR) and conditional value at risk (CVaR). Among these risk measures, CVaR has been gaining popularity due to both theoretical and computational advantages [36, 38]. However, there are two main obstacles: (i) most of the previous works focus on risk-neutral or static risk level in the single-agent settings, ignoring the randomness of reward and the temporal structure of agents' trajectories [8, 47, 24, 18]; (ii) many methods use risk measures over Q values for policy execution without getting the risk measure values used in policy optimization in temporal difference (TD) learning, which causes the global value factorization on expected individual values to have sub-optimal behaviours in MARL.

In this paper, we propose RMIX, a novel cooperative risk-sensitive MARL method. Specifically, our contributions are in three folds: (i) We first learn the return distributions of individuals by using Dirac Delta functions in order to analytically calculate CVaR for decentralized execution. The resulting CVaR values at each time step are used as policies for each agent via $\arg\max$ operation; (ii) We then propose a dynamic risk level predictor for CVaR calculation to handle the temporal nature of stochastic outcomes as well as tune the risk level during executions. The dynamic risk level predictor measures the discrepancy between the embedding of current individual return distributions and the embedding of historical return distributions. The dynamic risk levels are agent-specific and observation-wise; (iii) As our method focuses on optimizing the CVaR policies via CTDE, we finally optimize CVaR policies with CVaR values as target estimators in TD error via centralized training and CVaR values are used as auxiliary local rewards to update local return distributions via Quantile Regression loss. These also allow our method to achieve temporally extended exploration and enhanced temporal coordination, which are keys to solving complex multi-agent tasks. Empirically, we show that RMIX outperforms many state-of-the-art methods on various multi-agent risk-sensitive navigation scenarios and challenging StarCraft II cooperative tasks, demonstrating enhanced coordination and revealing improved sample efficiency.

## 2 Preliminaries and Related Works

**Dec-POMDP.** A fully cooperative MARL problem can be described as a *decentralised partially observable Markov decision process* (Dec-POMDP) [29] which can be formulated as a tuple $\mathcal{M} = \langle \mathcal{S}, \mathcal{U}, \mathcal{P}, R, \Upsilon, O, \mathcal{N}, \gamma \rangle$, where $s \in \mathcal{S}$ denotes the state of the environment. Each agent $i \in \mathcal{N} := \{1, ..., N\}$ chooses an action $u_i \in \mathcal{U}$ at each time step, giving rise to a joint action vector, $\boldsymbol{u} := [u_i]_{i=1}^N \in \mathcal{U}^N$. $\mathcal{P}(\boldsymbol{s'}|\boldsymbol{s}, \boldsymbol{u}) : \mathcal{S} \times \mathcal{U}^N \times \mathcal{S} \mapsto \mathcal{P}(\mathcal{S})$ is a Markovian transition function. Every agent shares the same joint reward function $R(\boldsymbol{s}, \boldsymbol{u}) : \mathcal{S} \times \mathcal{U}^N \mapsto \mathcal{R}$, and $\gamma \in [0, 1)$ is the discount factor. Due to *partial observability*, each agent has individual partial observation $\upsilon \in \Upsilon$, according to the observation function $O(\boldsymbol{s}, i) : \mathcal{S} \times \mathcal{N} \mapsto \Upsilon$. Each agent learns its own policy $\pi_i(u_i|\tau_i) : \mathcal{T} \times \mathcal{U} \mapsto [0, 1]$ given its action-observation history $\tau_i \in \mathcal{T} := (\Upsilon \times \mathcal{U})$.

**CVaR.** CVaR is a coherent risk measure and enjoys computational properties [36] that are derived for loss distributions in discreet decision-making in finance. It gains popularity in various engineering and finance applications. As shown in Fig. 1, CVaR is the expectation of values that are less equal than the $\alpha$-quantile value of the distribution over returns. Formally, let $X \in \mathcal{X}$ be a bounded random variable with cumulative distribution function $F(x) = \mathscr{P}[X \leq x]$ and the inverse CDF is $F^{-1}(u) = \inf\{x : F(x) \geq u\}$. CVaR at level $\alpha \in (0, 1]$ of a random variable $X$ is then defined as $\text{CVaR}_\alpha(X) := \sup_\nu \{\nu - \frac{1}{\alpha}\mathbb{E}[(\nu - X)^+]\}$ [37] when $X$ is a discrete random variable. Correspondingly, $\text{CVaR}_\alpha(X) = \mathbb{E}_{X \sim F}[X | X \leq F^{-1}(\alpha)]$

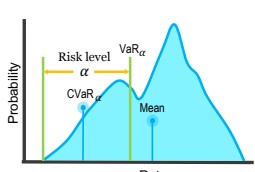

Figure 1: CVaR

[1] when $X$ is a continuous random variable. The $\alpha$-percentile value is value at risk (VaR). For ease of notation, we write CVaR as $\text{CVaR}_\alpha(F)$.

**Risk-sensitive RL.** Risk-sensitive RL uses risk criteria over policy/value, which is a sub-field of the Safety RL [11]. Von Neumann et al. [52] proposed the expected utility theory where a decision policy behaves as though it is maximizing the expected value of some utility functions. The condition is satisfied when the decision policy is consistent and has a particular set of four axioms. This is the most pervasive notion of risk-sensitivity. A policy maximizing a linear utility function is called *risk-neutral*, whereas concave or convex utility functions give rise to *risk-averse* or *risk-seeking* policies, respectively. Many measures are used in RL such as CVaR [6, 8] and power formula [8]. However, few works have been done in MARL. Our work fills this gap.

**Related Works.** CTDE [30] has drawn enormous attention via training policies of each agent with access to global trajectories in a centralized way and executing actions given only the local observations of each agent in a decentralized way. However, current MARL methods [22, 10, 43, 35, 42] neglect the limited representation of agent values, thus failing to consider the problem of random cost underlying the nonstationarity of the environment, a.k.a risk-sensitive learning. Recent advances in distributional RL [2, 9] focus on learning distribution over returns. However, these works still focus on either risk-neutral settings or with static risk level in single-agent settings. Chow et al. [5] considered the mean-CVaR optimization problem in MDPs and proposed policy gradient with CVaR. Garcia et al. [11] presented a survey on safe RL, which initiated the research on utilizing risk measures in RL [45, 47, 15, 26, 18, 24]. Bodnar et al. [3] proposed a risk-aware RL algorithm based on QR-DQN and CVaR for real vision-based robotic grasping tasks. However, these works focus on single-agent settings. The merit of CVaR in optimization of MARL has yet to be investigated.

## 3 Methodology

In this section, we present our framework RMIX in Fig. 2. In the rest of this section, we first introduce the CVaR operator to analytically calculate the CVaR value with the modeled individual distribution of each agent in Sec. 3.1 and then propose the dynamic risk level predictor to alleviate time-consistency issue in Sec. 3.2. Finally, we provide the details of centralized training of RMIX in Sec. 3.3. All proofs are provided in Appendix.

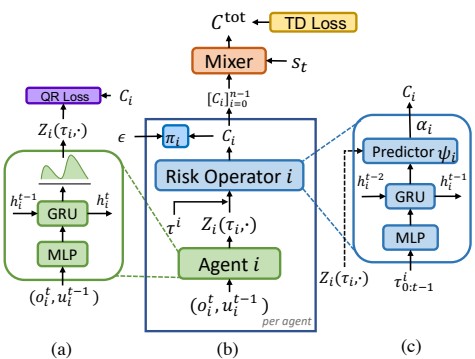

(a) (b) (c)

Figure 2: Our framework (dotted arrow indicates that gradients are blocked during training). (a) Agent's policy network. (b) The overall architecture (agent network and mixer). (c) Risk operator. Each agent $i$ applies an individual risk operator $\Pi_{\alpha_i}$ on its return distribution $Z_i(\cdot, \cdot)$ to calculate $C_i(\cdot, \cdot, \cdot)$ for execution given risk level $\alpha_i$ predicted by the dynamic risk level predictor $\psi_i$. $\{C_i(\cdot, \cdot, \cdot)\}_{i=1}^N$ are fed into the mixer for centralized training. Further introduction of the design of the agent, the risk operator and the mixer network can be found in the following sections.

### 3.1 CVaR of Return Distribution

In this section, we describe how we estimate the CVaR value. The value of CVaR can be either estimated through sampling or computed from the parameterized return distribution [36]. However, the sampling method is usually computationally expensive [47]. Therefore, we let each agent learn a return distribution parameterized by a mixture of Dirac Delta ($\delta$) functions [2], which is demonstrated to be highly expressive and computationally efficient [2]. By following [2], we define the parameterized return distribution of each agent $i$ at time step $t$ as:

$$Z_i^t(\tau_i, u_i^{t-1}) = \sum_{j=1}^{M} \mathscr{P}_j(\tau_i, u_i^{t-1}) \delta_j(\tau_i, u_i^{t-1}) \tag{1}$$

---

[2]The Dirac Delta is a *Generalized function* in the theory of distributions and not a function given the properties of it. We use the name *Dirac Delta function* by convention.

where $M$ is the number of Dirac Delta functions. $\delta_j(\tau_i, u_i^{t-1})$ is the $j$-th Dirac Delta function and indicates the estimated return which can be parameterized by neural networks in practice. $\mathscr{P}_j(\tau_i, u_i^{t-1})$ is the corresponding probability of the estimated return given local observations and actions. $\tau_i$ and $u_i^{t-1}$ are trajectories (up to that timestep) and actions of agent $i$, respectively. With the individual return distribution $Z_i^t(\tau_i, u_i^{t-1}) \in \mathcal{Z}$ and cumulative distribution function (CDF) $F_{Z_i(\tau_i, u_i^{t-1})}$, we define the CVaR operator $\Pi_{\alpha_i}$, at a risk level $\alpha_i$ ($\alpha_i \in (0,1]$ and $i \in \mathcal{A}$), over return as[3] $C_i^t(\tau_i, u_i^{t-1}, \alpha_i) = \Pi_{\alpha_i^t} Z_i^t(\tau_i, u_i^{t-1}) := \text{CVaR}_{\alpha_i^t}(F_{Z_i^t(\tau_i, u_i^{t-1})})$ where $C \in \mathcal{C}$. As we use CVaR on return distributions, it corresponds to risk-neutrality (expectation, $\alpha_i = 1$) and indicates the improving degree of risk-aversion ($\alpha_i \to 0$). $\text{CVaR}_{\alpha_i}$ can be estimated in a nonparametric way given ordering of Dirac Delta functions $\{\delta_j\}_{j=1}^M$ [19] by leveraging the individual distribution:

$$\text{CVaR}_{\alpha_i} = \sum\nolimits_{j=1}^M \mathscr{P}_j \delta_j \mathbf{1}\left\{\delta_j \leq \hat{v}_{M,\alpha_i}\right\}, \tag{2}$$

where $\mathbf{1}\{\cdot\}$ is the indicator function and $\hat{v}_{M,\alpha_i}$ is the $\alpha_i$-quantile of the individual distribution. This is a closed-form formulation and can be easily implemented in practice. The optimal action of agent $i$ can be calculated via $\arg\max_{u_i} C_i(u_i|\tau_i, u_i^{t-1}, \alpha_i)$. We will introduce it in detail in Sec. 3.2.

## 3.2 Risk Level Predictor

The values of risk levels, i.e., $\alpha_i$, $i \in \mathcal{A}$, are important for the agents to make decisions. Most of the previous works take a fixed value of risk level and do not take into account any temporal structure of agents' trajectories, which is hard to tune the best risk level and may impede centralized training in the evolving multi-agent environments. Therefore, we propose the dynamic risk level predictor, which determines the risk levels of agents by explicitly taking into account the temporal nature of the stochastic outcomes, to alleviate time-consistency issue [38, 17] and stabilize the centralized training. Specifically, we represent the risk operator $\Pi_\alpha$ by a deep neural network, which calculates the CVaR value with predicted dynamic risk level $\alpha$ over the return distribution. We illustrate how $\psi_i$ works with agent $i$ for CVaR calculation in practice in Fig. 3 in the following.

**Calculating the Risk Level.** We conduct the inner product to measure the discrepancy between the embedding of individual return distribution $Z_i$ and the embedding of the historical return distribution $\tilde{Z}_i$ for agent $i$. We discretize the risk level range into $K$ even ranges for the purpose of computing. The $k$-th dynamic risk level $\alpha_i^k$ is output from $\psi_i$ and the probability of $\alpha_i^k$ is defined as:

$$\mathscr{P}(\alpha_i^k) = \frac{\exp(\langle f_{\text{emb}}(Z_i)^k, f_{\text{emb}}(\tilde{Z}_i)^k\rangle)}{\sum_{k'=0}^{K-1} \exp(\langle f_{\text{emb}}(Z_i)^{k'}, f_{\text{emb}}(\tilde{Z}_i)^{k'}\rangle)}. \tag{3}$$

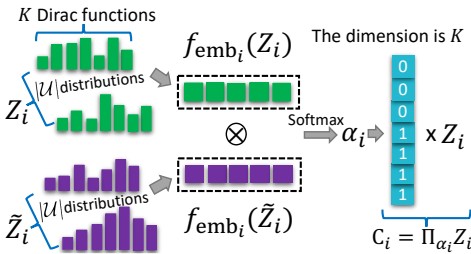

Figure 3: Risk level predictor $\psi_i$.

where $\tilde{Z}_i$ is modeled by GRU [7] with agent $i$'s past trajectory $\phi_i(\tau_i^{0:t-1}, u_i^{t-1})$ as input. Then we get the $k \in [1, \dots, K]$ with the maximal probability by $\arg\max$ and normalize it into $(0, 1]$, thus $\alpha_i = k/K$. The prediction risk level $\alpha_i$ is a scalar value and it is converted into a $K$-dimensional mask vector where the first $\lfloor \alpha_i \times K \rfloor$ items are one and the rest are zero. This mask vector is used to calculate the CVaR value (Eqn. 2) of each action-return distribution that contains $K$ Dirac functions. Finally, we obtain $C_i$ and the policy $\pi_i$ as illustrated in Fig. 2. During training, $f_{\text{emb}_i}$ updates its weights and the gradients of $f_{\text{emb}_i}$ are blocked (the dotted arrow in Fig. 2) in order to prevent changing the weights of the network of agent $i$. The dynamic risk level predictors allow agents to determine the risk level dynamically based on historical return distributions and it is a hyperparameter (e.g., $\alpha$) tuning strategy as well.

## 3.3 Centralized Training

We train RMIX via addressing the two challenging issues: credit assignment and local return distribution updating. The pseudo code of RMIX is in Algorithm 1. Agents observe observations and output actions in Line 6-8. In Line 10-13, agents' trajectories and states are stored in the replay buffer. Weights of the agent network and the mixing network are updated as shown in Line 15-22.

---

[3]We will omit $t$ in the rest of the paper for notation brevity.

**Algorithm 1:** RMIX

---

**1** **Input**: initialize parameters $\bar{\theta}$ and $\theta$ of the
    network and the target network of agents, risk
    operator, monotonic mixing network and
    replay buffer $\mathcal{D}$;
**2** **for** $e \in \{1, \ldots, MAX\_EPISODE\}$ **do**
**3**  **while** *EPISODE_NOT_DONE* **do**
**4**   Observe the global state $\boldsymbol{s}^t$;
**5**   **for** *agent* $i \in \{1, \ldots, N\}$ **do**
**6**    Observe $o_i^t$ and get action $u_i^{t-1}$ ;
**7**    Predict the risk level $\alpha_i$ (Eqn. 3);
**8**    Calculate CVaR values (Eqn. 2);
**9**    Get the action $u_i^t$;
**10**   Concatenate $u_i^t, i \in [1, .., N]$ into $\mathbf{u}_t$;
**11**   Execute $\boldsymbol{u}_i^t$ into environment;
**12**   Receive $r^t$ and observe a new state $\boldsymbol{s}'$;
**13**   Store $(\boldsymbol{s}^t, \{o_i^t\}_{i=1}^N, \boldsymbol{u}^t, r^t, \boldsymbol{s}')$ into $\mathcal{D}$;
**14**   **if** *UPDATE* **then**
**15**    Sample a min-batch $\mathcal{D}'$ from $\mathcal{D}$;
**16**    For each sample in $\mathcal{D}'$, calculate
       CVaR value $C_i$ (Eqn. 2);
**17**    Concatenate CVaR values
       $\{[\{C_i^1\}_{i=1}^N]_1, \ldots, [\{C_i^{|\mathcal{D}'|}\}_{i=1}^N]_{|\mathcal{D}'|}\}$;
**18**    For each $[\{C_i^j\}_{i=1}^N]_{0, j \in [1, \ldots, |\mathcal{D}'|]}$,
       calculate $C_j^{\text{tot}}$ via the mixing
       network;
**19**    Update $\theta$ by minimizing the TD
       loss (Eqn. 6);
**20**    **if** *UPDATE* **then**
**21**     Update $Z_i^t$ (Eqn. 7);
**22**   Update $\bar{\theta}$: $\bar{\theta} \leftarrow \theta$;
**23** **Output**: A well-trained policy for each agent.

---

**Loss Function.** As there is only a global reward signal and agents have no access to individuals' reward, we first utilize the monotonic mixing network ($f_m$) from QMIX to do credit assignment. $f_m$ enforces a monotonicity constraint on the relationship between $C^{\text{tot}}$ and each $C_i$ for RMIX:

$$\frac{\partial C^{\text{tot}}}{\partial C_i} \geq 0, \forall i \in \{1, 2, \ldots, N\}, \qquad (4)$$

where $C^{\text{tot}} = f_m(C_1(\cdot, \cdot, \cdot), \cdots, C_N(\cdot, \cdot, \cdot))$ and $C_i(\tau_i, u_i, \alpha_i)$ is the individual CVaR value of agent $i$. To ease the confusion, the $C^{\text{tot}}$ is not the global CVaR value as modeling global return distribution as well as local distribution are challenging with the credit assignment issue in Dec-POMDP problems, and the risk level values are locally decided and used during training. Then, to maximize the CVaR value of each agent, we define the risk-sensitive Bellman operator $\mathcal{T}$:

$$\mathcal{T}C^{\text{tot}}(\boldsymbol{s}, \boldsymbol{u}) := \mathbb{E}[R(\boldsymbol{s}, \boldsymbol{u}) + \gamma \max_{\boldsymbol{u}'} C^{\text{tot}}(\boldsymbol{s}', \boldsymbol{u}')] \qquad (5)$$

$\mathcal{T}$ operates on the $C^{\text{tot}}$ and the reward, which can be proved to be a contracting operation, as shown in Proposition 1.

**Proposition 1.** $\mathcal{T} : \mathcal{C} \mapsto \mathcal{C}$ *is a $\gamma$-contraction.*

*Proof.* It can be proved via contract mapping by integrating Lemma 3 in [6] and Eqn. 5. We provide the proof in Appendix. □

Therefore, we can leverage the TD learning [44] to train RMIX. Following the CTDE paradigm, we define our TD loss:

$$\mathcal{L}_\Pi(\theta) := \mathbb{E}_{\mathcal{D}' \sim \mathcal{D}}\left[(y_t^{\text{tot}} - C^{\text{tot}}(\boldsymbol{s}_t, \boldsymbol{u}_t))^2\right] \qquad (6)$$

where $y_t^{\text{tot}} = \left(r_t + \gamma \max_{\boldsymbol{u}'} C_{\bar{\theta}}^{\text{tot}}(\boldsymbol{s}_{t+1}, \boldsymbol{u}')\right)$. $\theta$ is the parameters of $C^{\text{tot}}$ which can be modeled by a deep neural network and $\bar{\theta}$ indicates the parameters of the target network which is periodically copied from $\theta$ for stabilizing training [28].

**Local Return Distribution Learning.** The CVaR estimation relies on accurately updating the local return distribution and the update is non-trivial. However, unlike many deep learning [12] and distributional RL methods [2, 9] where the label and local reward signals are accessible, in our problem, the exact rewards for each agent are unknown, which is very common in real world problems. To address this issue, we first consider CVaR values as dummy rewards of each agent due to its property of modeling the potential loss of return and then leverage the Quantile Regression (QR) loss used in Distributional RL [9] to explicitly update the local distribution decentally. More concretely, QR aims to estimate the quantiles of the return distribution by minimizing the quantile regression loss between $Z_i(\tau_i, u_i)$ and its target distribution $\hat{Z}_i(\tau_i, u_i) = C_i(\tau_i, u_i, \alpha_i) + \gamma Z_i(\tau_i', u_i')$. Formally, the quantile distribution is represented by a set of quantiles $\tau_j = \frac{j}{M}$ where $j \in [1, \ldots, M]$, and the quantile regression loss for Q network is defined in Eqn. 7,

$$\mathcal{L}_{QR} = \frac{1}{N} \sum_{i=1}^N \sum_{j=1}^M \mathbb{E}_{\hat{Z}_i \sim Z_i}[\rho_{\tau_j}(\hat{Z}_i - Z_i)] \quad (7) \qquad \mathcal{L}_\kappa(\nu) = \begin{cases} \frac{1}{2}\nu^2, & \text{if} |\nu| \geq \kappa, \\ \kappa(|\nu| - \frac{1}{2}\kappa), & \text{otherwise.} \end{cases} \qquad (8)$$

where $\rho_\tau(\nu) = \nu(\tau - \mathbf{1}\{\nu < 0\})$. To eliminate cuspid in $\rho_\tau$ which could limit performance when using non-linear function approximation, quantile Huber loss is used as the loss function. The quantile Huber loss is defined as $\rho_\tau(\nu) = \mathcal{L}_\kappa(\nu)|\tau - \mathbf{1}\{\nu < 0\}|$ where $\mathcal{L}_\kappa(\nu)$ is defined in Eqn. 8.

**Training.** Finally, we train RMIX in an end-to-end manner where each agent shares a single agent network and a risk predictor network to solve the lazy-agent issue [43]. $\psi_i$ is trained together with

the agent network via the loss defined in Eqn. 6. During training, $f_{\mathrm{emb}_i}$ updates its weights while the gradients of $f_{\mathrm{emb}_i}$ are blocked in order to prevent changing the weights of the return distribution in agent $i$. In fact, agents only use CVaR values for execution and the risk level predictor only predicts the $\alpha$; thus, the increased network capacity is mainly from the local return distribution and the CVaR operator. Our framework is flexible and can be easily used in many cooperative MARL methods.

## 4  Theoretical Analysis

Insightfully, our proposed method can be categorized into an overestimation reduction perspective which has been investigated in single-agent domain [48, 14, 20, 4]. Intuitively, during minimizing $\mathcal{L}_\Pi(\theta)$ and policy execution, we can consider CVaR implementation as calculating the mean over k-minimum $\delta$ values of $Z_i$. It motivates us to analyse our method's overestimation reduction property.

In single-agent cases, the overestimation bias occurs since the target $\max_{a'} Q\left(s_{t+1}, a'\right)$ is used in the Q-learning update. In multi-agent scenarios, for example StarCraft II, the primary goal for each agent is to survive (maintain positive health values) and win the game. Overestimation on high return values might lead to agents suffering defeat early-on in the game.

Formally, in MARL, we characterize the relation between the estimation error, the in-target minimization parameter $\alpha$ and the number of Dirac functions, $M$, which consist of the return distribution. We follow the theoretical framework introduced in [48] and extended in [4]. More concretely, let $C^{\mathrm{tot}}(\boldsymbol{s}, \boldsymbol{u}) - Q(\boldsymbol{s}, \boldsymbol{u})$ be the pre-update estimation bias for the output $C^{\mathrm{tot}}$ with the chosen individual CVaR values, where $Q(\boldsymbol{s}, \boldsymbol{u})$ is the ground-truth Q-value. We are interested in how the bias changes after an update, and how this change is affected by risk level $\alpha$. The post-update estimation bias, which is the difference between two different targets, can be defined as:

$$\Psi_{\boldsymbol{\alpha}} \triangleq r + \gamma \max_{\boldsymbol{u}'} C^{\mathrm{tot}}\left(\boldsymbol{s}', \boldsymbol{u}'\right) - (r + \gamma \max_{\boldsymbol{u}'} Q^{\mathrm{tot}}(\boldsymbol{s}', \boldsymbol{u}')) = \gamma(\max_{\boldsymbol{u}'} C^{\mathrm{tot}}(\boldsymbol{s}', \boldsymbol{u}') - \max_{\boldsymbol{u}'} Q^{\mathrm{tot}}(\boldsymbol{s}', \boldsymbol{u}'))$$

where $\boldsymbol{\alpha} = \{\alpha_i\}_{i=1}^N$ and $Q\left(\boldsymbol{s}', \boldsymbol{u}'\right)$ is output of the centralized mixing network with individuals' $Q_i$ as input. Note that due to the zero-mean assumption, the expected pre-update estimation bias is $\mathbb{E}[C^{\mathrm{tot}}(\boldsymbol{s}, \boldsymbol{u}) - Q(\boldsymbol{s}, \boldsymbol{u})] = 0$ by following [48] and [20]. Thus if $\mathbb{E}[\Psi_\alpha] > 0$, the expected post-update bias is positive and there is a tendency for over-estimation accumulation; and if $\mathbb{E}[\Psi_\alpha] < 0$, there is a tendency for under-estimation accumulation.

**Theorem 1.** *We summarize the following properties:*
*(1) Given $\boldsymbol{\alpha}_1$ and $\boldsymbol{\alpha}_2$ where $\alpha$ values in each set are identical, $\mathbb{E}[\Psi_{\boldsymbol{\alpha}_1}] \leq \mathbb{E}[\Psi_{\boldsymbol{\alpha}_2}]$ for $0 < \alpha_1 \leq \alpha_2 \leq 1$, $\alpha_1 \in \boldsymbol{\alpha}_1$ and $\alpha_2 \in \boldsymbol{\alpha}_2$.*
*(2) $\exists \alpha_i \in (0, 1]$ and $\boldsymbol{\alpha} = \{\alpha_i\}_{i=1}^N$, $\mathbb{E}[\Psi_{\boldsymbol{\alpha}}] < 0$.*

Theorem 1 implies that we can control the $\mathbb{E}[\Psi_{\boldsymbol{\alpha}}]$, bringing it from above zero (overestimation) to under zero (underestimation) by decreasing $\alpha$. Thus, we can control the post-update bias with the risk level $\alpha$ and boost the RMIX training on scenarios where overestimation can lead to failure of cooperation. In the next section, we will present the empirical results.

## 5  Experiments

We empirically evaluate our method on multi-agent risk-sensitive navigation scenarios and StarCraft II (SCII) cooperative scenarios. Especially, we are interested in the robust cooperation between agents and agents' learned risk-sensitive policies in complex cooperative scenarios.

### 5.1  Experiment Setup

**MACN.** We customize the cliff walking environment [44] in single-agent domain and develop Multi-Agent Cliff Navigation (MACN) for multi-agent risk-sensitive navigation. In MACN, there are two agents whose task is to complete the navigation from the starting position to the goal. At each time step, each agent observes an observation with a dimension of $3 \times 3$. Agents can take an action at each time step and the action set

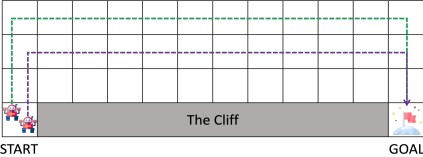

Figure 4: Multi-Agent Cliff Navigation.

is: $\{UP, DOWN, LEFT, RIGHT\}$. The two agents share the team reward. As depicted in Fig. 4, there are some regions that are dangerous and agents will be rewarded with a $-100$ reward when any agent steps into these regions, and consequently the episode ends. Agents will receive a $-1$ reward at each time step when they are at the safe region. When one agent reaches the goal, the agent will be rewarded with a $-0.5$ reward. If the two agents arrive at the goal at the same time, agents will be rewarded with a $0$ reward and the episode ends.

**SC II.** SMAC [39] is a challenging set of cooperative SCII maps for micromanagement MARL research. The enemy units are controlled by SCII built-in AI and each of the ally units is controlled by a learning agent. We present the results of our method and baselines on 6 scenarios: 1c3s5z, MMM2, 5m_vs_6m, 8m_vs_9m, 10m_vs_11m and corridor. The version of SCII we use in our experiments is 4.10. More information about MACN and SMAC can be found in Appendix B.

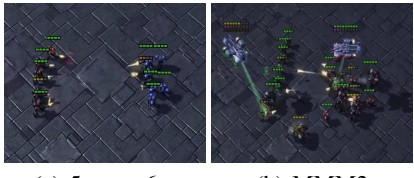

(a) 5m_vs_6m  (b) MMM2

Figure 5: SMAC scenarios.

**Baselines and training.** We compare three categories of MARL methods: (i) Value-based methods: IQL [46], VDN [43] , QMIX [35], QTRAN [42], WQMIX [33], QPLEX [53]; (ii) Policy gradient methods: COMA [10] and DOP [54]; (iii) Distributional RL-based method [23]. Among these methods, QPLEX (we use `num_circle=1` to train QPLEX for fair training at the sample level for each update) and WQMIX are state-of-the-art value-based MARL methods and DOP is state-of-the-art policy gradient MARL method. Additional introduction can be found in Appendix D. We implement our method on PyMARL [39] and use 5 random seeds to train each method. We carry out experiments on NVIDIA Tesla V100 GPU 16G and NVIDIA GeForce RTX 3090 24G. More training details can be found in Appendix C. Our code can be found at this link: `https://github.com/yetanotherpolicy/rmix`.

## 5.2 Experimental Results

In this subsection, we first showcase the learned risk-sensitive policies of RMIX in grid-world MACN scenarios and then present the improved complex coordination of RMIX in SCII scenarios.

### 5.2.1 Multi-Agent Cliff Navigation

We conduct experiments on two multi-agent cliff navigation scenarios to showcase the gained performances of RMIX and the learned risk-sensitive policy. MACN scenario 1 has a width of 7 and a height of 4 while MACN scenario 2 has a width of 12 and a height of 4. The baselines are QMIX, QPLEX and DOP, which are well received risk-neutral MARL methods. As shown

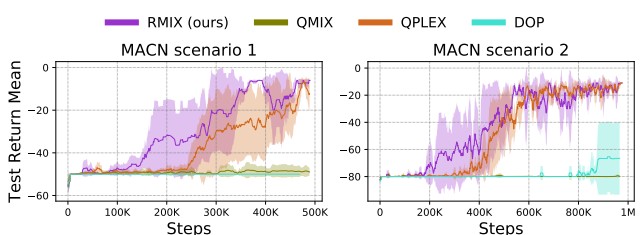

Figure 6: Test return on two MACN scenarios.

in Fig. 6, trained with risk-sensitive policies, RMIX demonstrates better sample efficiency compared with other risk-neutral, i.e., expected-Q, baseline methods in two MACN scenarios with high risk. QPLEX outperforms QMIX and DOP and shows converged performance due to the dueling network [55] and attention architecture [49]. In Sec.5.3, we illustrate experimental results on less risky MACN scenarios.

### 5.2.2 StarCraft II

**Comparison with value-based MARL baselines.** As depicted in Fig. 7, RMIX outperforms QMIX, QPLEX, WQMIX, IQL, QTRAN and VDN in all *asymmetric* (5m_vs_6m, 8m_vs_9m, 10m_vs_11m and corridor)/*symmetric* (MMM2 and 1c3s5z) and *homogeneous* (5m_vs_6m, 8m_vs_9m, 10m_vs_11m and corridor)/*heterogeneous* (1c3s5z and MMM2) scenarios, demonstrating its superiority on improving multi-gent coordination in complex scenarios. We can find that RMIX improves coordination in a sample efficient way via risk-sensitive policies. Intuitively, for

*asymmetric* scenarios, agents can be easily defeated by the opponents. As a consequence, coordination between agents is cautious in order to win the game, and the cooperative strategies in these scenarios should avoid massive casualties in the starting stage of the game.

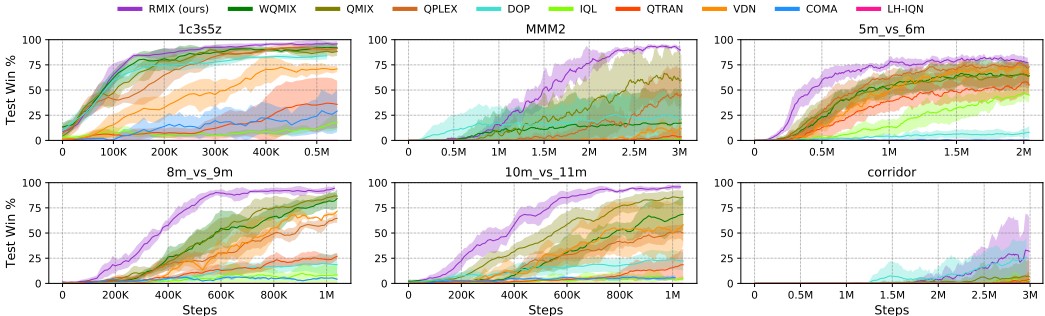

Figure 7: Test win rates for six scenarios.

**Comparison with policy gradient MARL baseline.** Our method also outperforms COMA and DOP as illustrated in Fig. 7. COMA aims to address multi-agent credit assignment issue by utilizing counterfactual values. DOP learns the centralized Q value for policy learning with a linear combination of individual Q values. The two methods are on-policy methods and are sample inefficient.

**Comparison with distributional/risk-sensitive MARL baseline.** Although there are few practical methods on risk-sensitive multi-agent reinforcement learning, we also conduct experiments to compare our method with LH-IQN [23], which is a risk-sensitive method built on a distributional RL method called implicit Quantile network (IQN). We can find that our method outperforms LH-IQN in many scenarios. The performance of LH-IQN is not good on StarCraft II scenarios because it is an independent MARL method and the IQN was used to guide the update of the Lenient Q-Network.

## 5.3 Ablations

RMIX consists of two components: the CVaR policies and the risk level predictor. The CVaR policies are different from vanilla Q values and the risk level predictor is proposed to model the temporal structure and as an $\alpha$-finding strategy for hyperparameter tuning. Our ablation studies serve to answer the following questions: **Q1:** *Can RMIX with static $\alpha$ also work?* **Q2:** *Can the risk level predictor learn $\alpha$ values and fast learn a good policy compared with RMIX with static $\alpha$?* **Q3:** *Can our framework be applied to other methods?* **Q4:** *Is our method robust to the randomness of rewards?* **Q5:** *How does our method perform in less risky scenarios?*

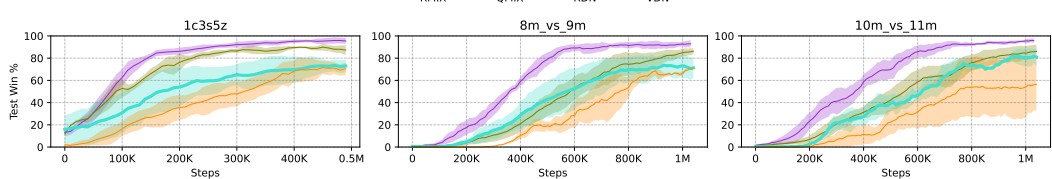

Figure 8: Test win rate of RMIX, RDN, VDN, QMIX.

To answer **Q1** and **Q2**, we conduct an ablation study by fixing the risk level in RMIX with the value of $\{0.1, 0.3, 0.5, 0.7, 1.0\}$ and compare with RMIX and QMIX on 1c3s5z (500K time steps) and 5m_vs_6m (2M time steps). As illustrated in Table 1, with static $\alpha$ values, RMIX is capable of learning good performance over QMIX, which demonstrates the benefits of learning risk-sensitive MARL policies in complex scenarios where the potential of loss should be taken into consideration in coordina-

Table 1: Test Win Rate.

|          | 1c3s5z            | 5m_vs_6m          |
|----------|-------------------|-------------------|
| RMIX     | $97.10 \pm 2.56\%$ | $80.01 \pm 5.57\%$ |
| RMIX_0.1 | $93.48 \pm 1.67\%$ | $60.65 \pm 6.07\%$ |
| RMIX_0.3 | $95.47 \pm 2.84\%$ | $69.16 \pm 4.74\%$ |
| RMIX_0.5 | $95.15 \pm 1.63\%$ | $70.75 \pm 4.73\%$ |
| RMIX_0.7 | $95.89 \pm 2.15\%$ | $74.88 \pm 8.48\%$ |
| RMIX_1.0 | $89.70 \pm 8.22\%$ | $77.56 \pm 4.11\%$ |
| QMIX     | $88.34 \pm 2.64\%$ | $64.75 \pm 8.69\%$ |

tion. With risk level predictor, RMIX outperforms RMIX with static $\alpha$ values, illustrating that agents have captured the temporal features of scenarios and possessed the $\alpha$ value tuning merit.

We answer **Q3** to show that our proposed method is applicable in other value-based MARL methods. We then apply additivity of individual CVaR values to represent the global CVaR value as $C^{\mathrm{tot}}(\boldsymbol{\tau}, \boldsymbol{u}) = C_1(\tau_1, u_1, \alpha_1) + \cdots + C_N(\tau_N, u_N, \alpha_N)$. Following the training of RMIX, we name this method **R**isk **D**ecomposition **N**etwork (**RDN**). We use experiment setup of VDN and train RDN on 3 SMAC scenarios. With CVaR policies, RDN outperforms VDN on 10m_vs_11m and converges faster than VDN on 1c3s5z and 8m_vs_9m, as depicted in Fig. 8, demonstrating that our framework can be applied to VDN and outperforms VDN.

We answer **Q4** by directly injecting Gaussian noise ($\mathcal{N}(0, \delta)$, $\delta \in \{0.1, 0.3, 0.5, 0.7, 1.0\}$) to the reward function. The reward function in SMAC is deterministic, however, each agent owns stochastic policy which makes the nonstationarity of the environment and leads to the randomness of reward, especially at the early training stage where agents explore. It has also been claimed by distributional RL [2, 8] in single-agent setting. Results in Fig. 9 show that RMIX is robust to random noise in the reward function.

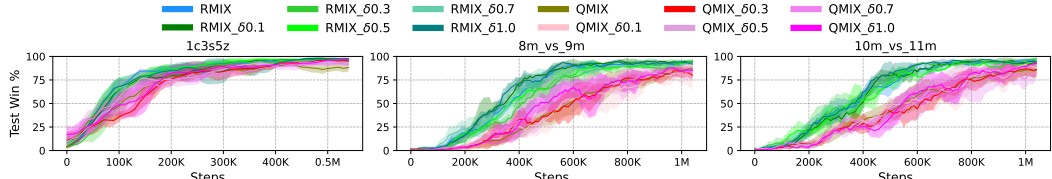

Figure 9: RMIX vs QMIX with reward noise.

We answer **Q5** by conducting experiments on MACN scenario with low risk. We select MACN scenario 1 as the evaluation scenario where the width is 7 and height is 4. In order to make the scenario less risky, we change the punishment of stepping into the dangerous region, a.k.a cliff, from the value of -100 to -50. We present the averaged test return value of RMIX, QMIX, QPLEX and DOP in Fig.10. We can find that in scenario with low risk, unlike results in Fig. 6, RMIX shows comparable converged performance with QPLEX. The variance of the averaged test return of RMIX is larger than that of QPLEX. We can also find the improved performance of QMIX in this low risk scenario. We can conclude that in low risk scenarios, RMIX can also show comparable performance compared with risk-neutral MARL methods.

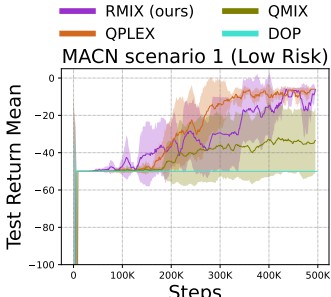

Figure 10: Test averaged return in MACN scenario 1.

## 5.4 Results Analysis

We are interested in finding if the risk level predictor can predict temporal risk levels. We use the trained model of RMIX on corridor and run the model to collect one episode data including game replay, states, actions, rewards and $\alpha$ values. In Fig. 11, the first row shows the reward of each time step and the second row shows the $\alpha$ value each agent predicts per time step. We can find that $\alpha$ values vary in different phases within the episode. Agent 3 dies at the time step 20 and its risk level is decreasing as illustrated in Fig. 11, demonstrating risk-averse behaviours. The cuspid (time step 45-53) of agent 0's $\alpha$ curve indicates that agent 0 is confronting opponents alone and later its teammates come over and help him to survive. Consequently, the $\alpha$ value increases to 1 (risk neutral).

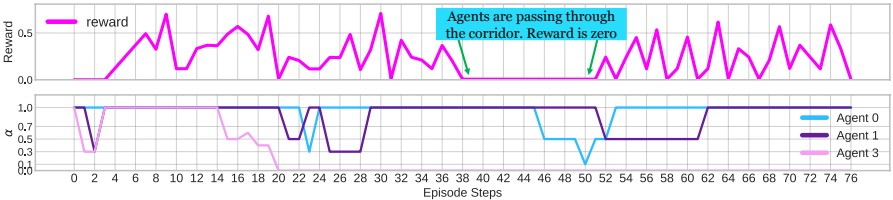

Figure 11: Reward and risk level on corridor.

We showcase how agents apply the risk level to select actions. We collect the executed actions, the predicted risk levels and the local distributions of agent 1 (Zealot) of time step 28 and 74. Rewards

and the risk levels have already been shown in Fig. 11. As depicted in Fig. 12, there are 35 Dirac delta atoms (for better visualization) of the local distributions as shown in the heat map. The x-axes denote each Dirac function. The y-axes stand for available actions at current time step for agent 1 while the size of the action space is 30. At time step 28, as shown in Fig. 12 (a), the predicted risk level is 0.3. We sort values in the heat map of each action in ascending order and then apply Eqn. 2 to get CVaR values of each action. Values in white cells in the heat map (Fig. 12 (a)) are excluded while calculating the CVaR value. At time step 28, because many opponents are attacking agent 1, risk level is 0.3 (means agent 1 is going to encounter potential reward loss) and agent 1's action is $\mathrm{Attack}$. In Fig. 12 (b), at time step 74, as agents are going to win the game, agent 1 does not need to get involved in the combat, agent 1's risk level is 1.0 and it takes $a_3$, i.e., $\mathrm{MoveSouth}$, to get out of the battlefield to avoid being killed by the opponent. The heat map illustrates that all the return values are used for CVaR calculation. Readers can refer to Appendix E.2 and click the link in the supplemental file to watch the video for more details.

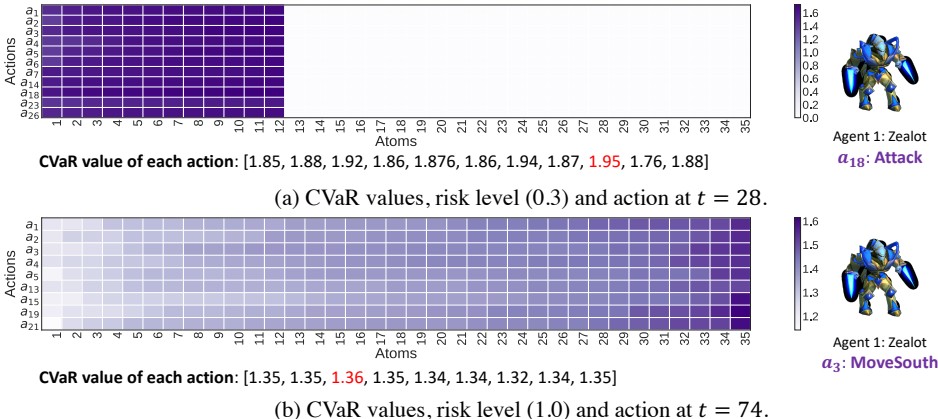

**CVaR value of each action**: [1.85, 1.88, 1.92, 1.86, 1.876, 1.86, 1.94, 1.87, 1.95, 1.76, 1.88]

(a) CVaR values, risk level (0.3) and action at $t = 28$.

**CVaR value of each action**: [1.35, 1.35, 1.36, 1.35, 1.34, 1.34, 1.32, 1.34, 1.35]

(b) CVaR values, risk level (1.0) and action at $t = 74$.

Figure 12: Two examples on CVaR calculation and action selection.

## 6 Conclusion and Future Work

In this paper, we propose RMIX, a novel and practical MARL method with CVaR over the learned distributions of individuals' Q values as risk-sensitive policies for cooperative agents. Empirically, we show that our method outperforms baseline methods on many challenging StarCraft II tasks, reaching convincing performance and enhanced coordination as well as improved sample efficiency.

Risk-sensitive policy learning is vital for many real-world multi-agent applications especially in risky tasks, for example autopilot vehicles and finance portfolio management. For the future work, better risk measurement together with accurate spatial-temporal trajectory representation can be investigated. Also, learning to model other agents' risk levels and reach consensus with communication can be another direction for enhancing multi-agent coordination. Our method is built on value-based MARL method and devising risk-sensitive policy gradient MARL methods is our future direction too.

## 7 Social Impact and Limitations

Our work provides a practical framework for future risk-sensitive MARL research. In addition to the algorithmic contribution, our method could help the development of various real-world applications where risk-sensitive decision-making is desperately needed. For example, the driving policy should be risk-sensitive to any potential accidents and uncertainties in automatic driving scenarios. Besides training risk-sensitive policies for automatic driving, the proposed method is applicable for multi-robot rescue where imprudent decision can lead to the failure of the mission.

The proposed method trains MARL policies in a risk-sensitive manner with CVaR. However, the learned policy does not guarantee the exact safety bound. We adopt the Quantile-Regression [9] method to estimate the local return distribution accurately and efficiently train the CVaR values, which is not scalable for scenarios that have many agents and large action space. The aforementioned limitations motivate us to develop novel risk-sensitive MARL methods in the future.

## Acknowledgments and Disclosure of Funding

We thank Yanchen Deng and Tabish Rashid for helpful comments and suggestions on the draft. This work was supported by the National Research Foundation, Singapore under its AI Singapore Programme (AISG Award No: AISG-RP-2019-0013), National Satellite of Excellence in Trustworthy Software Systems (Award No: NSOE-TSS2019- 01). We gratefully acknowledge the GPU tech support from Jianxiong (Terry) Yin, NVAITC (NVIDIA AI Tech Center) for our research.

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
