# OpenReview forum: "RMIX: Learning Risk-Sensitive Policies for Cooperative Reinforcement Learning Agents"
_NeurIPS.cc/2021/Conference — NeurIPS 2021 Poster_

### Official Review · Reviewer_CWF4 · 2021-07-12

**Rating:** 7
**Confidence:** 4

**Summary:**

This work takes risk-sensitive learning to cooperative multiagent learning. It first, learns the conditional value at risk of the return distribution then, a second component is a risk-level predictor which dynamically adapts the risk. Lastly, the paper builds on QMIX to deal with the credit assignment problem since there' is a single global reward.

The problem the authors are tackling are very hard and I see the benefits of their approach.
I think the experimental results are interesting and informative, however the paper needs some clarifications to be more understandable to a bigger audience.



**Limitations And Societal Impact:**

No discussion from the authors.

**Main Review:**

Originality
Risk-sensitive approaches have been proposed mainly in single-agent RL. Within MARL, the area seems less explored.

Clarity
The paper has a lot of content in fact, I think removing some parts might help the readability, I have some small suggestions to improve.

- The theoretical result is interesting but I don't see it as the major contribution, actually I think that section can be shortened.
- You added the pseudocode of the algorithm to the main paper, however, I suggest to add accompanying text to it.
- I know about the lazy agent issue but I'm not sure I understand your comment: "we train RMIX in an end-to-end manner where each agent shares a single agent network and a risk predictor network to solve the lazy-agent issue".
- I'm not sureI agree with:  "the overall structure of QPLEX is more complex and benefits much from increased model capacities than RMIX."
- Figure 7 is quite small to see the differences
- Is Figure 4 really necessary to have?
- Extend the caption in Figure 3

Quality

I see the benefits, but I would like to know what are the disadvantages of modeling the return distribution with a mixture of Dirac delta functions? Is this the same approach as Distributional RL with implicit quantile networks ?

Why there is a need to compute the discrepancy wrt to the historical distribution? (Section 3.2)
- Why the historical information is useful?


Significance

- The experimental results look solid and the ablation results show the benefits of the proposed components.

Minor:

"discreet decision"
"It gains popularity in various engineering and finance applications. " ??
"However, few works have been done in MARL and it cannot be easily extended." ??


**Time Spent Reviewing:**

4

---

> ### Author Response · Authors · 2021-08-10
> **Responses to Reviewer CWF4**
>
> We thank the reviewer’s very useful comments and suggestions on our submission. We summarize the reviewer’s questions and present our responses below.
>
> ---
>
> **Q1: (1) I think theoretical result section can be shortened. (2) I suggest to add accompanying text to the pseudocode. (3) Figure 7 is quite small to see the differences. (4) Is Figure 4 really necessary? (5) "discreet decision" (6) "It gains popularity in various engineering and finance applications. " ?? (7) "However, few works have been done in MARL and it cannot be easily extended." ??**
>
> **A1:**
>
> (1), (2), (3), (6) and (7): We will update our paper and enlarge Fig. 7.
>
> (4): We followed QMIX, COMA and presented Fig. 4.
>
> (5): “discreet” means cautious or prudent. It looks similar to “discrete”.
>
> ---
>
> **Q2: I know about the lazy agent issue but I'm not sure I understand your comment.**
>
> **A2:** In that sentence, we wanted to point out that we also used parameter-sharing, which is widely used in VDN [1], QMIX and other MARL methods to alleviate the lazy agent problem.
>
> ---
>
> **Q3: I'm not sure I agree with: "the overall structure of QPLEX is more complex and benefits much from increased model capacities than RMIX."**
>
> **A3:** QPLEX [2] was claimed as the SOTA in SMAC. It got the improved performance in SCII with three main components:
>
> (1) Dueling network [2] (Dueling network outperforms DQN, DDQN) (Sec. 3.2 in [2]);
>
> (2) Weights generated for $V_i (\tau_i)$ and $A_i (\tau_i, a_i)$ (Eqn. 7 in [2]) with agent's trajectory $\tau_i$ as input;
>
> (3) Weights generated by Attention network on $A_i(\tau, a_i)$ (Eqn. 9 and 10 in [2]) with global trajectory $\tau$, which provides more information for estimating the weights.
>
> (4) QPLEX uses multi-run training where the sampled trajectories are used to conduct the loss minimization for multiple times for each update.
>
> However, RMIX has simpler architecture. There are no attention networks and we do not use multi-run training. The CVaR is a scalar value for policy execution and the risk level predictor only outputs a scalar value for CVaR calculation. The QR in our framework is used to update the local return distribution for each agent. The RL framework of RMIX is simpler than that of QPLEX.
>
> ---
>
> **Q4:  Why there is a need to compute the discrepancy wrt to the historical distribution and why the historical information is useful?**
>
> **A4:** Because we want to predict the risk level for each agent. Using the agent’s past return distributions with the agent’s current return distribution can stabilize the risk level predictor.
>
> ---
>
> **Q5: What are the disadvantages of modeling the return distribution with a mixture of Dirac delta functions? Is this the same approach as Distributional RL with implicit quantile networks ?**
>
> **A5:** (1) Modelling the return distribution with a mixture Dirac delta functions makes the training speed slightly slower [4] as there are many parameters to update. We summarize the training time of some Q value-based methods in the following. In the table below, we can see that RMIX is slightly slower in some scenarios. The training time is acceptable.
>
> |Scenarios |	RMIX |      QMIX | QPLEX |	 VDN  |	     IQL | WQMIX |
> |:-------------:|:-------------:|:-------------:|:-------------:|:-------------:|:-------------:|:-------------:|
> |1c3s5z	  |12 hours	 | 9 hours	 |12 hours  |7 hours	  |6.5 hours | 9 hours.     |
> |MMM2	      |22 hours	| 20 hours |1 day 5 hours	|18 hours	|19 hours	|20 hours|
> |5m_vs_6m	|20 hours	| 18 hours | 19 hours | 18 hours | 9 hours | 13 hours |
> |8m_vs_9m	| 8 hours	| 8 hours	| 12.5 hours	| 8 hours	| 8 hours	| 8 hours |
> |10m_vs_11m	| 9 hours	| 8 hours	| 11 hours | 8 hours	| 7 hours	| 9 hours |
> |corridor	| 1day 14 hours	| 24 hours	| 2 day 1 hours	| 21 hours	| 22 hours	| 24 hours|
>
> We also present the memory usage (given the current size of the replay buffer) for the training of each method(exclude COMA and DOP, which are on-policy methods without using experience replay) on scenarios of the SCII domain in SMAC (Table 3 in the Appendix).
>
> |Scenario  |Memory Usage (GB)|
> |:--------------:|:-------------------:|
> |5m_vs_6m 	|	  3    |
> |8m_vs_9m 	|	 4.9 |
> |10m_vs_11m 	|       7.1   |
> |1c3s5z 		|	8.6 |
> |MMM2 		|       10.8 |
> |Corridor 		|        14.4 |
>
>
>
> (2) The Dirac delta functions are widely used for modelling probability functions. Bellemare et al. [4] also used it to represent return distributions. Later QR-DQN and IQN (implicit Quantile Networks) also used Dirac delta functions to represent distributions.
>
> ----
>
> **Q6: The paper needs some clarifications to be more understandable to a bigger audience.**
>
> **A6:** Our target audiences are researchers in MARL.
>
> ---
>
> **References:**
>
> [1] Sunehag, Peter, et al. "Value-decomposition networks for cooperative multi-agent learning." arXiv preprint arXiv:1706.05296 (2017).
>
> [2] Wang, Jianhao, et al. "QPLEX: Duplex Dueling Multi-Agent Q-Learning." International Conference on Learning Representations. 2020.
>
> [3] Wang, Ziyu, et al. "Dueling network architectures for deep reinforcement learning." International conference on machine learning. 2016.
>
> [4] Bellemare, Marc G., Will Dabney, and Rémi Munos. "A distributional perspective on reinforcement learning." International Conference on Machine Learning. PMLR, 2017.

---

> ### Author Response · Authors · 2021-08-27
> **Dear Reviewer CWF4, did our responses address your questions?**
>
> Dear Reviewer CWF4,
>
> As the response system will be closed soon within one week. We thank you again for your comments. We hope our detailed responses could address your questions. More questions on our paper are always welcomed! If there are no more questions and we will appreciate it if you can kindly raise the score.
>
> Sincerely yours,
>
> Authors of Paper828

---

> > ### Comment · Reviewer_CWF4 · 2021-08-31
> > **Raising score**
> >
> > Thank you for the detailed answers, I think now I understand more about your method and many of my doubts were answered. I'll raise my score accordingly.

---

> > > ### Author Response · Authors · 2021-09-01
> > > **Thank You**
> > >
> > > Dear Reviewer CWF4,
> > >
> > > We are glad to hear that our response helped you understand our method. Thank you very much for raising the score.

---

### Official Review · Reviewer_iywX · 2021-07-12

**Rating:** 7
**Confidence:** 3

**Summary:**

This paper presents a multi-agent RL algorithm based on CVaR. In the proposed algorithm, the return distribution is learned as in QR-DQN, and CVaR is estimated using the trained model. The centralized training for multi-agent settings is based on QMIX. In the proposed algorithm, Q-values in QMIX is replaced with CVaR.  The proposed method is evaluated with several scenarios in StarCraftII, and the proposed methods outperformed baseline methods.

**Limitations And Societal Impact:**

Although the policies are trained in a risk-sensitive manner, the proposed algorithm does not guarantee the safety during the training phase. I recommend authors to clarify this point in the paper in order not to confuse/mislead audiences, who are not machine learning experts.

**Main Review:**

Strong points:
-	The proposed method was rigorously evaluated and outperformed baseline methods.

Weak points:
-	The contribution seems incremental: Risk-aware RL algorithm based on QR-DQN and CVaR was proposed in [1], although it is not cited. The proposed algorithm RMIX is basically the combination of QMIX and Q2-Opt proposed in [1].

I appreciate the authors' efforts for performing comparison with baseline methods and ablation study. On the other hand, the use of the QR-DQN for computing CVaR is not novel.  In the context of robotic manipulation, Bodnar et al. proposed an algorithm called Q2-Opt, which is a risk-aware RL algorithm based on QR-DQN and CVaR[1]. Adding [1] to the reference is recommended. The proposed algorithm RMIX seems the combination of QMIX and Q2-Opt. Namely, the proposed algorithm can be obtained by replacing Q values in QMIX with the CVaR in Q2-Opt.

However, the risk predictor is not proposed in Q2-Opt, and it is the contribution of this paper. The proposed algorithm is rigorously evaluated and outperformed baseline methods. If Q2-Opt paper [1] is added to the related work and the authors clearly state that a risk-aware RL algorithm based on QR-DQN and CVaR has been proposed in [1], I think the paper can meet the standard of NeurIPS.

[1] Cristian Bodnar, Adrian Li, Karol Hausman, Peter Pastor, Mrinal Kalakrishnan.  Quantile QT-Opt for Risk-Aware Vision-Based Robotic Grasping. R:SS, 2020.

=== comments after the author response ===

I went through other reviewers' comments and the author response. I think that the authors addressed the concerns raised by reviewers.
I increase the score from 6 to 7.

**Time Spent Reviewing:**

3

---

> ### Author Response · Authors · 2021-08-10
> **Responses to Reviewer iywX**
>
> We thank the reviewer’s insightful comments on our work.
>
> ---
>
> **Q1: Adding Bodnar et al. (2020) to the reference is recommended. If Q2-Opt paper [1] is added to the related work and the authors clearly state that a risk-aware RL algorithm based on QR-DQN and CVaR has been proposed in Bodnar et al. (2020), I think the paper can meet the standard of NeurIPS.**
>
> **A1:** We noticed the contributions of Q2-Opt [1]. Bodnar et al. (2020) proposed a method based on distributional RL for real vision-based robotic grasping tasks. In our final version, we will clearly state that a risk-aware RL algorithm based on QR-DQN and CVaR has been proposed in Bodnar et al. (2020). Recent successes of risk-sensitive RL in single-agent domains motivated us to apply it to multi-agent cooperative RL where multiple agents act cooperatively to complete some tasks.
>
> ---
>
> **References**:
>
> [1] Cristian Bodnar, Adrian Li, Karol Hausman, Peter Pastor, Mrinal Kalakrishnan. Quantile QT-Opt for Risk-Aware Vision-Based Robotic Grasping. R:SS, 2020.

---

> > ### Comment · Reviewer_iywX · 2021-08-18
> > **Thank you for the response**
> >
> > Thank you for the response. I went through other reviewers' comments and authors' response.
> > I think that the concerns raised by the reviewers are reasonably addressed. I will raise the score if other reviewers do not raise a significant concern.

---

> > > ### Author Response · Authors · 2021-08-19
> > > **Thank you for your feedback**
> > >
> > > Dear reviewer, thank you for your feedback on our responses.

---

> > > ### Author Response · Authors · 2021-08-28
> > > **To Reviewer iywX**
> > >
> > > Dear Reviewer iywX,
> > >
> > > As the response system will be closed within a few days and reviewers 5UXh, CWF4, and QUdu did not respond to our responses. We will appreciate it if you can kindly consider raising the score. Thank you.
> > >
> > > Sincerely yours,
> > >
> > > Authors of Paper828

---

> > > > ### Author Response · Authors · 2021-09-01
> > > > **Thank You**
> > > >
> > > > Dear Reviewer iywX,
> > > >
> > > > Thank you very much for raising the score.

---

### Official Review · Reviewer_QUdu · 2021-07-16

**Rating:** 6
**Confidence:** 3

**Summary:**

This is an interesting article introducing risk-averse distributional RL ideas into the multi-agent setting in a new and it seems useful way. It builds on methods like Q-Mix and VDN but replaces normal returns with CVar_alpha, the expected return for the worst alpha-percentile outcomes. They also introduces a method for picking alpha different for different timesteps and different agents.

**Limitations And Societal Impact:**

ok

**Main Review:**

This is an interesting article introducing risk-averse distributional RL ideas into the multi-agent setting in a new and it seems useful way. It builds on methods like Q-Mix and VDN but replaces normal returns with CVar_alpha, the expected return for the worst alpha-percentile outcomes. They also introduces a method for picking alpha different for different timesteps and different agents, before mixing them into the total, which slightly confuses me and I would like to see more conceptual discussion on what that choice is really optimized for and why it is a good idea, though their ablation study shows that it does better than any fixed choice so something smart is probably done but I have not really understood the what and why it is good.

The empirical evaluation first shows superior performance in the SCII micromanagement tasks compared to start-of-the-art algorithms which is interesting, while reminiscent of what was seen with risk-averse distributional rl for the atari suite of tasks when adding this to DQN-based agents. There have been various hypotheses why this was the case including that it just usefully takes more information into account akin to auxiliary tasks. The author goes on to show the usefulness for the tasks with extra inserted noise, which is more directly testing the aimed for robustness, though Q-mix was doing too badly in this either (Fig 7).

It seems like a new useful iteration of MARL algorithms for the team reward setting, VDN -> Q-Mix -> R-Mix, where they also show that adding it straighter to VDN giving RDN is also clearly useful.

I would mainly like to better understand the “prediction” of alpha better the thinking behind how this is done.

The contribution builds on things already developed for the single agent setting, but requires some amount of innovation for the marl setting.


**Time Spent Reviewing:**

3-4 hours

---

> ### Author Response · Authors · 2021-08-10
> **Responses to Reviewer QUdu**
>
> We thank the reviewer’s helpful feedback on our submission.
>
> ---
>
> **Q1: Give more conceptual discussions on what that choice is really optimized for and why it is a good idea. … I would mainly like to better understand the “prediction” of alpha better the thinking behind how this is done.**
>
> **A1:** The motivation of picking alpha differently for different timesteps and different agents is that each agent is partial observable and at time step, each agent observes different observations and the risk level can be different. Using static risk levels can gain acceptable performance. However, at different time steps, the potential loss of return can vary. For example, when there are few teammates nearby during the combat with enemies, the potential loss of return can be very high and the static risk level cannot reflect the current “risk” of the agent correctly. With the risk level predictor, we can predict the current risk level with the agent’s current return distribution and its previous return distributions. By taking into account such temporal nature of the stochastic outcomes, we can also alleviate the time-consistency issue and stabilize the centralized training.
>
> We also provided an example in Sec. 5.4 and Appendix E.2 in our paper to illustrate the risk level predictor. It shows that the agent is able to learn different risk levels given different observations. The video is available in the folder of supplementary materials.

---

### Official Review · Reviewer_5UXh · 2021-07-30

**Rating:** 7
**Confidence:** 3

**Summary:**

The authors present RMIX, which is a value-based MARL method incorporating a risk evaluator to prevent overly optimistic actions in teamwork among agents. In particular, the CVaR metric is chosen for its computational advantage and theoretically sound foundation. Experiments in the SMAC environment show better win rates than do recent baselines.

**Limitations And Societal Impact:**

I would up my rating if the authors would kindly come up with some toy examples (one low-risk setting showing little to no performance difference between RMIX and risk-unaware algorithms and one high-risk setting showing larger performance difference between them). Something like the CliffWalk example illustrating the difference between Q-learning and SARSA would do. Or, any equivalently valid explanation about interpreting the intrinsic risk levels of a task and the corresponding effects in RMIX would be much appreciated.

**Main Review:**

Originality: Contribution of the work is clear. There is a well-defined niche in the literature among MARL and safety RL. Authors appropriately position RMIX in the intersection of value-based MARL and risk-sensitive RL. The non-stationarity challenge addressed in the paper is unique to the multi-agent setting; a simple application of the single-agent CVaR to the multi-agent setting would not work. However, the said challenge are not specific to RMIX, and thus the (partial) solutions (utilizing recurrent models and CTDE) also borrow from existing literature. It would have been even better if the non-trivialities arising from the multi-agent nature of the risk-sensitive RL problem were highlighted.

Quality: The paper was easy to follow. Figure 2 delivers the overall structure clearly. Evaluation includes some of the more recent works and a thorough ablation study and discussion, without which the complexity of the SMAC tasks might have clouded the effectiveness of the risk sensitivity.

Clarity: The theory that supports the work feels rather fragmented. For example, the appendix states Proposition 2 as key to understanding and proving Theorem 1. What compelled the authors to include contraction mapping theorem, which in my opinion is the relatively "standard" one, and leave out Proposition 2 instead? Please consider unifying the theory overarching the work.

Significance: RMIX is being compared with the baselines that it should be compared with, plus some other family of MARL algorithms as well. It is shown to excel in most scenarios. A brief explanation on how QPLEX beat RMIX in the 8m_vs_9m task would be informative. In fact, most SMAC tasks are rather complex, so simpler didactic tasks would probably better illustrate the effectiveness of RMIX. For instance, if risk-unaware algorithms (such as QPLEX) outperformed better than RMIX, would it be correct to say that the 8m_vs_9m task carries little to no risk? There is some guesswork (on my part) involved when attempting to decide whether a situation carries much risk or not. Simpler, albeit toy-ish, tasks may be built to better compare two distinctively constructed scenarios of varying risk levels.

**Time Spent Reviewing:**

5

---

> ### Author Response · Authors · 2021-08-10
> **Responses to Reviewer 5UXh**
>
> We thank the reviewer’s valuable feedback on our work. We summarize the reviewer’s questions and present our responses below.
>
> ---
>
> **Q1: It would have been even better if the non-trivialities arising from the multi-agent nature of the risk-sensitive RL problem were highlighted.**
>
> **A1:** We believe that the main challenge in risk-sensitive MARL is that each agent acts in a decentralized way and can have varying potential loss of returns. It imposes challenges for complex multi-agent coordination for risk-sensitive MARL, rendering the time-consistency problem.
>
> ---
>
> **Q2: What compelled the authors to include contraction mapping theorem, which in my opinion is the relatively "standard" one, and leave out Proposition 2 instead?**
>
> **A2:** In centralized training with CTDE, with agents’ risk-sensitive values fed into the mixing network to estimate the global $C_{tot}$, it is necessary to clarify that it is gamma-contraction in MARL, which has been clarified in some single-agent risk-sensitive RL methods. The Proposition 2 was proposed for our method’s overestimation reduction property. We will unify the results of the theoretical analysis to make them clearer to readers.
>
> ---
>
> **Q3: A brief explanation on how QPLEX beat RMIX in the 8m_vs_9m task would be informative.**
>
> **A3:** We think the main reasons are that 8m_vs_9m is easier and QPLEX owns many advanced components.
>
> (1) Compared with other scenarios, 8m_vs_9m (easy) is simpler than 5m_vs_6m (super hard), MMM2 (super hard) and corridor (super hard) where RMIX outperforms QPLEX;
>
> (2) QPLEX [1] was claimed as the SOTA in SMAC. It got the increased performance in SCII with three main components: (i) Dueling network [2] (Dueling network outperforms DQN, DDQN) (see Sec. 3.2 in [1]); (ii) Weights generated for $V_i (\tau_i)$ and $A_i (\tau_i, a_i)$ (Eqn. 7 in [1]) with agent's trajectory $\tau_i$ as input; (iii) Weights generated by Attention network on $A_i(\tau, a_i)$ (Eqn. 9 and 10 in [1]). Besides that, QPLEX uses multi-run training, namely, the sampled trajectories are used to conduct the loss minimization for multiple times for each update. However, RMIX has simpler architecture. There are no attention networks and we do not use multi-run training. The CVaR is a scalar value for policy execution and the risk level predictor only outputs a scalar value for CVaR calculation. The RL framework of RMIX is simpler than that of QPLEX;
>
> (3) Cautious actions may impede the performance in simpler scenarios. In 8m_vs_9m, QPLEX converges slightly faster than RMIX before step 600K, after that, RMIX outperforms QPLEX and eventually both methods converge nearly to the same resulting performance.
>
> ---
>
> **Q4: Toy examples (one low-risk setting showing little to no performance difference between RMIX and risk-unaware algorithms and one high-risk setting showing the larger performance difference between them). Or, any equivalently valid explanation about interpreting the intrinsic risk levels of a task and the corresponding effects in RMIX would be much appreciated.**
>
> **A4:**
> (1) We conduct experiments on toy Multi-Agent Cliff Navigation (MACN) scenarios to demonstrate the risk-sensitive policies learned by RMIX. MACN is a customized environment for multi-agent navigation with risk, which is adapted from the Cliff Walking environment [3] for single-agent RL research.
>
> In MACN, two agents have the task of completing the navigation from the starting position to the goal. At each time step, each agent observes an observation with a dimension of $3\times3$. Agents take an action at each time step and the action set is: $\{UP,  DOWN, LEFT, RIGHT\}$. The two agents share the team reward. There is a cliff to the south of the scenario. Agents will be rewarded with a $-100$ reward when any agent steps into the cliff, and consequently, the episode ends. Agents will receive a $-1$ reward at each time step when they are at the safe region. When one agent reaches the goal, the agents will be rewarded with $-0.5$. If the two agents arrive at the goal simultaneously, agents will be rewarded with a $0$ reward and the episode ends. The objective of the agents is to maximize the accumulated rewards in each episode. An episode ends after 20 steps or agents step into the cliff.
>
> The experimental results demonstrate that our risk-sensitive MARL method RMIX outperforms risk-neutral MARL methods even in simple toy scenarios with risk. RMIX converges much faster and gains more stable performance than QMIX and QPLEX in the scenario with high risk. While in the scenario with low risk, RMIX has comparable performance with QPLEX. QMIX performs more stable in the scenario with high risk.
>
> We also provide an anonymous (no identifying information) link to show our results: https://sites.google.com/view/rmix2021. We will put the experimental results into the final version.
>
> (2) In many multi-agent cooperative tasks, the risk level is changing over time (as shown in Fig. 8 in our paper) because different observations at different time steps can have a different potential loss of returns (risk) for each agent, the CVaR policy is used for decentralized execution, the risk level predictor was proposed for alleviating the time-consistency problem, namely, we want the agent can use time-observation related risk level for execution. In centralized training, on one hand, we use centralized training to train the policies with the credit assignment network. On the other hand, we use centralized training to update the risk level prediction network with the shared trajectories of each agent.
>
>
>
>
> ---
>
>
> **References:**
>
> [1] Wang, Jianhao, et al. "QPLEX: Duplex Dueling Multi-Agent Q-Learning." International Conference on Learning Representations. 2020.
>
> [2] Wang, Ziyu, et al. "Dueling network architectures for deep reinforcement learning." International conference on machine learning. 2016.
>
> [3] Sutton, Richard S., and Andrew G. Barto. Reinforcement learning: An introduction. MIT press, 2018.

---

> > ### Comment · Reviewer_5UXh · 2021-09-01
> > **Thank you for the response**
> >
> > Thank you, authors.
> >
> > I find the answers provided convincing. I don't think there is any major concern left unaddressed.
> >
> > I am leaning more strongly towards acceptance, raising the score by 1 point from 6 to 7.

---

> > > ### Author Response · Authors · 2021-09-01
> > > **Thank You**
> > >
> > > Dear Review 5UXh,
> > >
> > > We are glad to hear that our previous responses addressed your concerns. Thank you for raising the score.

---

### Author Response · Authors · 2021-08-17
**Dear reviewers, did our responses address your questions?**

We thank again all the reviewers for their constructive and valuable comments. We really appreciate the positive comments made by reviewers who recognised our contribution to MARL.

We hope our responses, including experiments on toy examples, could address the questions of all the reviewers. More discussions and suggestions on our paper are also always welcomed!

Sincerely yours,

Authors of Paper828

---

### Decision · Program_Chairs · 2021-09-27

**Decision:**

Accept (Poster)

**Comment:**

All reviewers felt the author rebuttals satisfactorily addressed their concerns and three reviewers increased their scores as a result.  Overall, there was a unanimous decision to accept.  The authors are strongly encouraged to incorporate review comments and rebuttal discussion into their final revision.